# Study on the Influence Mechanism of the Key Active Structure of Coal Molecules on Spontaneous Combustion Characteristics Based on Extraction Technology

Jun Guo [1,2], Yunfei Wu [1,*], Yin Liu [1,3], Guobin Cai [1] , Dailin Li [1] and Yan Jin [1,4]

1 College of Safety Science and Engineering, Xi'an University of Science and Technology, Xi'an 710054, China; guojun@xust.edu.cn (J.G.); liuyin@xust.edu.cn (Y.L.); 19220214086@stu.xust.edu.cn (G.C.); 22220226172@stu.xust.edu.cn (D.L.); 19220214099@stu.xust.edu.cn (Y.J.)
2 Key Laboratory of Western Mine and Hazard Prevention, Ministry of Education of China, Xi'an 710054, China
3 College of Energy Engineering, Xi'an University of Science and Technology, No. 58, Yanta Mid. Rd., Xi'an 710054, China
4 Taizhou Petroleum Branch, Sinopec Sales Co., Ltd., Taizhou 318000, China
* Correspondence: 23220226169@stu.xust.edu.cn

**Abstract:** The molecular structure of coal is complex, and the existing research methods are limited, so it is difficult to clarify its influence mechanism on the spontaneous-combustion characteristics of coal. In this paper, the previous extraction, FTIR, TPR, TG-DSC and other experimental results are combined to analyze the extraction weakening effect and the correlation analysis of the spontaneous-combustion characteristic parameters of raffinate coal. The results show that extraction can destroy the connection bond of coal molecules, change the content of dominant active groups in the coal spontaneous-combustion reaction, increase the lower limit of the key temperature nodes of coal spontaneous-combustion or extend the temperature range, resulting in an increase in the ignition-point temperature of coal and a decrease in coal quality. This paper will provide a theoretical basis for the study of the microscopic mechanism of coal spontaneous-combustion and then provide new ideas for the development of an active prevention and control technology for coal spontaneous-combustion.

**Keywords:** coal spontaneous-combustion; solvent extraction; key active structures; correlation analysis; weakening mechanism

## 1. Introduction

As the most abundant fossil fuel in the world [1,2], coal resources are accompanied by many safety problems in the process of mining. The harm of coal spontaneous-combustion exists in the whole process of coal mining, storage and transportation [3,4]. The accidents caused by coal spontaneous-combustion account for more than 90% of mine fire accidents [5,6], which cause a lot of waste of coal resources and hinders the safe, green and sustainable development of China's coal industry [7,8].

Coal is a complex mixture of organic matter and inorganic matter, and its physical and chemical properties are affected by its molecular structure and active groups [9–12]. Qiu et al. [13] found that the content of oxygen-containing functional groups in the material is positively correlated with the specific surface area. When the content of oxygen-containing functional groups is too high, the specific surface area will shrink due to the blockage of micropores and the collapse of pore walls. Dhakate et al. [14] found that the proportion of surface reactive functional group content will affect the strength of carbon fiber in the process of studying carbon fiber materials. In the process of evaluating Russian needle coke carbon materials, Rudko et al. [15] found that materials with different hydrocarbon compositions and sulfur content would lead to changes in the structure and properties of the products. Coal spontaneous combustion is the result of heat storage and combustion after the coal–oxygen composite reaction between the coal molecular active structure and

oxygen [16–18]. Previous studies have confirmed that the microstructures of the aliphatic hydrocarbon structure, aromatic hydrocarbon structure and oxygen-containing functional groups in coal molecules have different influence mechanisms and influence degrees in coal spontaneous combustion [19]. Therefore, controlling or eliminating the key active structure of coal molecules is an effective means to prevent and control coal spontaneous combustion. Solvent extraction technology can effectively strip the specific active structure of coal molecules and achieve the purpose of inhibiting coal spontaneous combustion [20]. Some scholars have successfully stripped the microscopic active structure of coal molecules by solvent extraction technology. Zhao et al. [21] found that the content of oxygen-containing functional groups such as phenols in coal samples dissolved by non-volatile ionic liquids was significantly reduced. Jin et al. [22] used diphenylamine to treat coal samples and found that the N-H in coal molecules was destroyed. Yao et al. [23] successfully stripped the aliphatic hydrocarbon structure in coal molecules using tetrahydrofuran combined with ultrasonic-assisted extraction technology. Barman et al. [24] reviewed the application of ultrasonic-assisted extraction technology in coal extraction and found that the content of functional groups on the surface of coal changed after a solvent combined with ultrasonic-assisted extraction.

On this basis, some scholars have studied the weakening mechanism of coal spontaneous combustion with changes in microstructure. Taraba et al. [25] used urea and other reagents to treat coal samples, and combined with a pulse flow calorimetry test, it was found that urea reduced the oxidation heat of coal at the chemical level, thus weakening the spontaneous-combustion activity of coal. Wang et al. [26] significantly delayed the ignition temperature of a test coal sample by adding dimethyl methylphosphonate (DMMP) material and determined by kinetic analysis that DMMP can also physically weaken the coal–oxygen composite reaction activity intensity in the low-temperature stage by blocking oxygen at the physical level. Onifade et al. [27] used gypsum powder to make antioxidant materials to treat the sponge coal of a South African mine. Through testing, gypsum materials can effectively reduce the oxygen consumption of coal and greatly reduce the spontaneous-combustion of coal. Deng et al. [28] found that when testing [BMIM][BF$_4$] materials the temperature threshold of the spontaneous-combustion reaction process interval of coal samples pretreated with ionic liquids was expanded, that is, the spontaneous-combustion of coal was weakened by delaying the characteristic temperature node of coal spontaneous combustion reaction acceleration. Zhang et al. [29] found that by reducing the content of the -OH structure, the quality and heat release of coal entering the combustion stage can be significantly reduced, and then the spontaneous-combustion of coal can be weakened. Slovák et al. [30] tested bituminous coal by using CaCl$_2$ and urea as a composite inhibitor and found that CaCl$_2$ and urea in the range of 100~250 °C will lead to a decrease in the heat released by coal oxidation. At the same time, CaCl$_2$ can inhibit the spontaneous-combustion of coal by increasing the activation energy in the process of coal spontaneous combustion.

The quantitative state (content and state) of the active structure of coal molecules is a key factor affecting the spontaneous-combustion characteristics of coal [31]. The above scholars have carried out research on the weakening mechanism of solvent extraction residue on the reactivity of coal spontaneous combustion and have achieved relevant results, but the influence of the key active structures on the characteristics of coal spontaneous combustion needs further study. Therefore, based on extraction technology, combined with an infrared test, temperature-programmed experiment and other means, this paper will test and analyze extracted coal and use the gray correlation method to analyze the correlation between the active group structure and the characteristic parameters of coal spontaneous combustion, to determine the weakening mechanism of the key active structure of coal molecules on the spontaneous-combustion characteristics. The research results provide research methods and methods for clarifying the microscopic reaction mechanism of coal spontaneous combustion and provide a theoretical basis for promoting the development of the active prevention and control of coal spontaneous combustion disasters in mines.

## 2. Experiment and Theoretical Analysis

### 2.1. Experiments of Extraction

In the experiment, 10 groups of extractants with 20% solute density were selected, and each group was configured with 5 L of solvent [32]. The extractants were as follows: cyclohexane + sodium dioctyl succinate sulfonate + anhydrous ethanol (CYC + AOT + AE), sodium dioctyl succinate sulfonate (AOT), cyclohexane (CYC), anhydrous ethanol (AE), cellulase (EG), tea polyphenols (TP), cellulase + tea polyphenols (EG + TP), tetrahydrofuran (THF), methanol (MT), n-hexane (CYH) and the raw coal control group (YM). During the extraction process, 200 g of samples with five particle sizes of 0~0.9, 0.9~3, 3~5, 5~7 and 7~10 mm was mixed into 1 kg of experimental coal samples, and the extraction solvent was added and fully mixed, followed by static extraction. After the extraction, the extracted coal samples were collected for drying treatment, and the coal samples were sealed and stored for later use.

### 2.2. Extraction of Residual Coal Group Content Test

A Vertex 70 V Fourier transform infrared spectrometer was used to test and determine the extraction effect of the group structure of the residual coal sample in each reagent group [32]. The experimental sample and dried potassium bromide powder (KBr) were mixed at a mass ratio of 1:150. After full grinding, 200 mg of the ground coal sample and KBr powder mixture was filled in a tableting metal mold. The mixed sample was placed on a tableting machine. Under a pressure of 15 Mpa, the mixed sample was made into an observation sheet and placed in a Fourier transform infrared spectrometer for experimental testing. The setting parameters were resolution 4 cm$^{-1}$, band $400^{-4}$~4000 cm$^{-1}$, scan 32 times and export test analysis data preservation.

### 2.3. Determination of the Macroscopic Characteristic Parameters of Coal Oxidation

2.3.1. Temperature-Programmed Experiment Test

The experimental system included five parts, an adiabatic oxidation device, gas supply system, temperature-tracking detection and control system, oxidation gas product analysis system and mass change detection system, and was connected with a ZDC 7 mine fire multi-parameter intelligent monitoring device [33]. During the test, 800 g of dry raffinate coal sample was placed in the coal sample tank for sealing, and the air pump was adjusted for preventilation. The initial temperature of the box was set at 40 °C, and the heating rate was 0.8 °C/min to start the test. During the heating process, the ZDC 7 monitoring device was used to monitor the central temperature of the coal sample and the outlet oxygen volume fraction at the key node temperature in real time, and the temperature of the box was collected at the same time. The intersection point of the temperature– time curve of the coal center and the temperature–time curve of the box is the temperature of the intersection point. The data index can indirectly characterize the spontaneous-combustion strength of the test coal sample.

2.3.2. Isothermal Temperature Difference Leading Test

The main structure of the experiment included five parts: an adiabatic oxidation device, gas flow detection device, flow control system, temperature detection system and parameter control system [33]. In the experiment, 80 g of dry extracted coal sample was placed in the coal sample tank for heating up. The leading temperature was set to 10 °C, and the heating rate was 0.8 °C/min. The gas generated in the coal sample tank was introduced into the gas chromatography analyzer every 10 °C to detect the gas composition. The gas type and volume fraction data at different temperatures were recorded until the test was completed at 180 °C. Through experiments, the formation law of gaseous products in the oxidation process of raffinate coal samples can be obtained, and then the spontaneous-combustion reactivity intensity of each group of raffinate coal samples can be reflected.

2.3.3. Thermal Analysis Combined Test (TG-DSC)

An STA 8000-Spectrum thermal analyzer was used in the experiment [34]. During the experiment, 10 mg of dry extracted coal samples was placed in the sample room. The preloading temperature was 30 °C, the termination temperature was 800 °C, the heating rate was 10 °C/min, the preloaded gas flow rate was 100 mL/min, nitrogen was used as the protective gas and 21% oxygen was used as the purging gas during the test. At the same time, the temperature of each instrument interface and transmission network pipeline was maintained at 200 °C during the heating process of the control program. The changes in the mass and airflow heat data of the test samples during the heating process were recorded. Through a data fitting processing analysis, the differences in the thermal stability and thermal effect characterization parameters in the process of the oxidation and spontaneous-combustion of each group of raffinate coal were determined.

*2.4. The Principle of Gray Correlation Analysis*

Gray correlation analysis is an analysis method based on the existing data to calculate the correlation degree between the reference sequence and the comparison sequence [29]. The correlation degree between the research object and the characteristic parameters can be determined by gray correlation analysis. The larger the correlation degree is, the greater the correlation between the research object and the characteristic parameters. The specific calculation steps are as follows:

1.  Determine the reference sequence (1) and the comparison sequence (2):

$$Y = Y(k)|k = 1, 2 \ldots n, \tag{1}$$

$$X_i = X_i(k)|k = 1, 2 \ldots n, i = 1, 2 \ldots m, \tag{2}$$

2.  The initial value method is used to make the data dimensionless.
3.  Calculate the correlation coefficient; the calculation formula is as follows (3):

$$\xi_i(k) = \frac{\min\limits_{i}\min\limits_{k}|y(k) - x_i(k)| + \rho\max\limits_{i}\max\limits_{k}|y(k) - x_i(k)|}{|y(k) - x_i(k)| + \rho\max\limits_{i}\max\limits_{k}|y(k) - x_i(k)|}, \tag{3}$$

4.  Calculate the correlation degree, that is, calculate the average value of the obtained correlation coefficient to obtain the correlation degree, as shown in Equation (4):

$$r_i = \frac{1}{n}\sum_{k=1}^{n} \xi_i(k), k = 1, 2, \ldots n, \tag{4}$$

In the formula, Y is the reference sequence, $X_i$ is the comparison sequence, $\xi_i$ is the correlation coefficient, $\rho$ is the resolution coefficient, usually taken as $\rho = 0.5$, and $r_i$ is the correlation degree.

*2.5. Analysis Process of the Influence Mechanism of the Active Structure on Coal Spontaneous-Combustion Based on Macro–Micro Correlation*

In this paper, the extraction experiment of coal samples was carried out based on extraction technology. In order to determine the influence mechanism of the active group structure on the reactivity of coal spontaneous combustion, a Fourier transform infrared spectroscopy test, thermal analysis combined test, temperature-programming experiment and isothermal temperature difference leading experiment test were carried out. By analyzing the differences in the characteristic data of the test and analysis results of each group of raffinate coal samples, the influence of different group structures on the actual coal spontaneous-combustion process is determined, and then the influence mechanism of the key active structure of coal molecules on the spontaneous-combustion characteristics is theoretically deduced. The specific technical roadmap is shown in Figure 1.

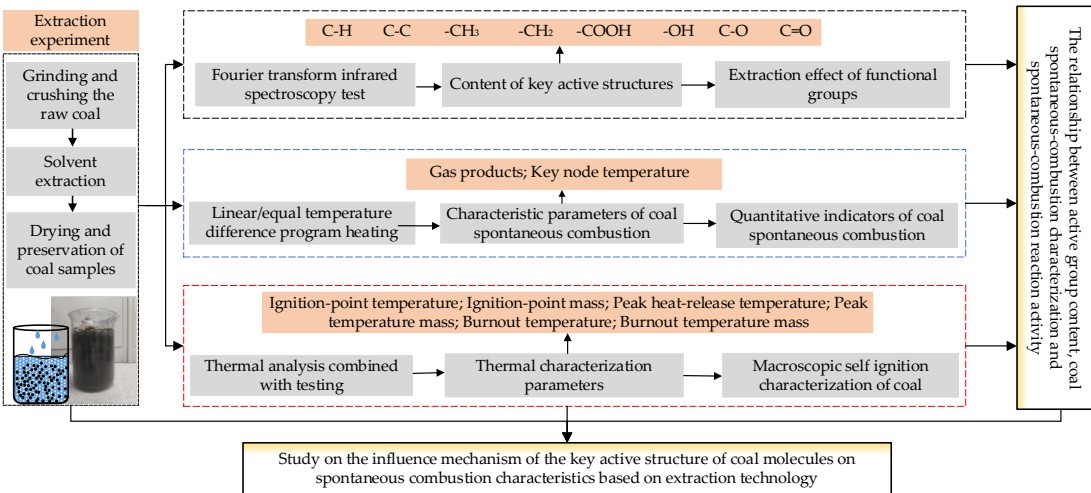

**Figure 1.** Technical roadmap.

## 3. Results and Analysis

In the previous study, the team carried out extraction experiments on Caojiatan coal samples and combined them with Fourier transform infrared spectroscopy to optimize the extraction solvent. Then, a temperature-programmed experiment was carried out on the extracted coal. By calculating the quantitative index of the spontaneous-combustion of extracted coal, the spontaneous-combustion weakening effect of coal samples treated with different extractants was obtained [32]. In addition, combined with the formation law of gas products and the change law of thermal physical parameters during the heating process of extracted coal, it was found that the temperature node of the spontaneous-combustion reaction interval of the coal sample treated by the extractant shifted, and the ignition-point temperature of the coal was delayed, resulting in a weakening of the spontaneous-combustion reaction activity of the coal [33,34].

However, the influence mechanism of the active structure on coal spontaneous combustion characteristics is not clear, and the quantitative characteristics between the active structure and coal spontaneous combustion characteristic parameters, gas products and thermophysical parameters are difficult to clarify. Therefore, this paper combines the experimental results and data of the previous team research to deeply analyze the problem.

### 3.1. Change in Active Group Content of Extracted Residual Coal

Based on the test and analysis process and calculation results of the previous raffinate coal experiment [32,33], the micro-group structure types of each group of raffinate coal samples were fitted according to the aromatic hydrocarbon structure, aliphatic hydrocarbon structure and oxygen-containing functional group structure. The absorption peaks were summarized, and the extraction effect of each extractant was analyzed. The infrared fitting peak area of the active structure of each raffinate coal group is shown in Table 1 and Figure 2 below.

Oxygen-containing functional groups are the most abundant in coal molecular groups and play an important role in the process of coal spontaneous combustion. The oxygen-containing functional groups represented by-OH play a leading role in the low-temperature oxidation stage of coal spontaneous combustion, and their content is positively correlated with the strength of coal spontaneous combustion. It can be seen from Figure 2 that regarding the correlation fitting peak area of each group of raffinate coal samples, the larger the fitting peak area, the worse the extraction effect. Therefore, the extraction effect of the oxygen-containing group structure from strong to weak is as follows: MT > EG > THF > AE > TP > CYC > EG + TP > CYC + AOT + AE > AOT > CYH > YM. Among them, the reagents with the best extraction effect of the oxygen-containing group structure were the

THF group, EG group and MT group, and the extraction rates were 13.9%, 14.7% and 19.1%, respectively.

**Table 1.** Infrared peak fitting results (dimensionless).

| Group | Active Structure Category | YM | THF | MT | CYC | CYH | C + A + A | AE | AOT | EG | TP | EG + TP |
|---|---|---|---|---|---|---|---|---|---|---|---|---|
| Aromatic | C-H | 108 | 77 | 91 | 56 | 42 | 46 | 54 | 51 | 73 | 45 | 40 |
| hydrocarbon | C=C | 185 | 199 | 157 | 190 | 204 | 201 | 166 | 211 | 173 | 179 | 172 |
| Aliphatic | -CH$_3$ | 52 | 43 | 15 | 49 | 29 | 40 | 54 | 130 | 58 | 65 | 45 |
| hydrocarbon | -CH$_2$ | 117 | 73 | 101 | 44 | 28 | 63 | 96 | 145 | 103 | 78 | 111 |
| Oxygen- | -COOH | 187 | 161 | 154 | 176 | 176 | 181 | 184 | 174 | 185 | 183 | 168 |
| containing | -OH | 435 | 388 | 310 | 389 | 463 | 428 | 357 | 480 | 357 | 367 | 425 |
| functional | C-O | 177 | 137 | 147 | 144 | 134 | 150 | 145 | 153 | 146 | 154 | 160 |
| groups | C=O | 66 | 59 | 89 | 75 | 93 | 75 | 63 | 56 | 50 | 50 | 50 |

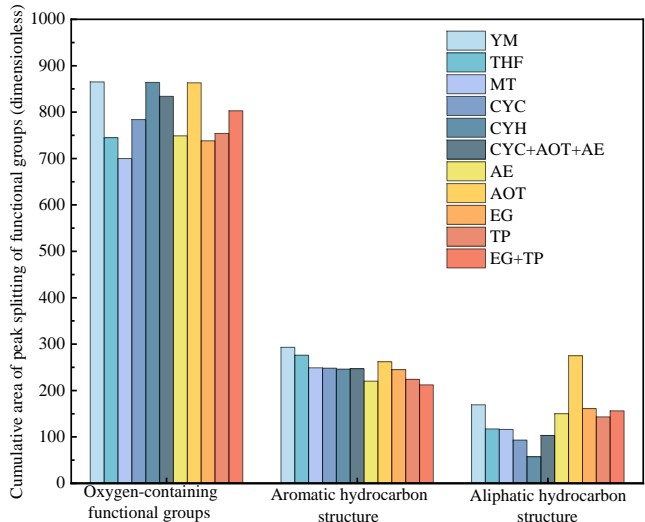

**Figure 2.** Comparison of the difference in group content under different extractants.

As the skeleton of coal molecules, the aromatic hydrocarbon structure is composed of a benzene ring, aromatic ring and so on. This kind of structure is very stable in coal molecules, and it is usually difficult to react with oxygen. It can be seen from Figure 2 that the order of the aromatic structure extraction effect from strong to weak is as follows: EG + TP > AE > TP > EG > CYH > CYC + AOT + AE > CYC > MT > AOT > THF > YM. Among them, the extraction residual coal group with the best extraction effect was the TP group, AE group and EG + TP compound group, and the extraction rate reached 23.5%, 24.9% and 27.6%, respectively.

The aliphatic hydrocarbon structure generally exists in coal molecules as aromatic ring side chains, and it is easy to react with oxygen during the coal spontaneous-combustion reaction. It can be seen from Figure 2 that the extraction effect of the aliphatic hydrocarbon structure from strong to weak is as follows: CYH > CYC > CYC + AOT + AE > MT > THF > TP > AE > EG + TP > EG > YM > AOT. Among them, the extraction residual coal group with the best extraction effect was the CYC + AOT + AE compound group, CYC group and CYH group, and the extraction rate reached 39.1%, 45.0% and 66.3%, respectively. The absorption peak area of aliphatic hydrocarbons in the AOT group increased. It is speculated that the AOT material contains a large amount of methyl structure, and the material is retained inside the coal pores during the extraction process, resulting in an increase in the area of infrared absorption peaks here. In addition, due to the physical and chemical properties of the AOT material itself, it is easy to form a reverse micelle layer wrapped on

the surface of the coal body during the extraction process, which blocks the dissolution of aliphatic hydrocarbon structures such as methyl and methylene, resulting in the partial stripping of aliphatic hydrocarbon structures. It is also retained in the coal pores, resulting in an increase in the absorption peak area of the aliphatic hydrocarbon structure of the AOT group.

### 3.2. Effect of Active Group Content on Coal Spontaneous Combustion

The results of previous studies have found that [33] the spontaneous-combustion characteristics of extracted coal samples have changed to varying degrees. The determination method of coal oxidation kinetics [35] was used to calculate the spontaneous-combustion characteristics of each group of extracted coal samples based on the measured center temperature of each group of extracted coal samples and the volume fraction of $O_2$ at key points. It was found that except for the enhancement of the spontaneous-combustion of coal in the AE group, the spontaneous-combustion reactivity of coal in other groups was weakened to varying degrees. Another characterization of the change in coal spontaneous-combustion is the change in the coal spontaneous-combustion process interval. Figure 3 shows the change in the spontaneous-combustion process interval of each extraction residual coal group. It can be seen from the figure that the characteristic temperature interval of the coal spontaneous-combustion process of each extraction residual coal group has been extended. At the same time, there is no phenomenon of a compound self-heating stage in the extraction residual coal of the AE group.

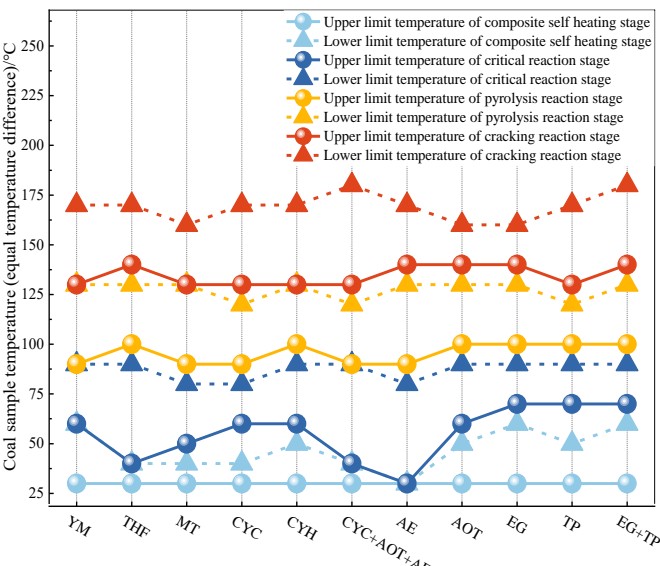

**Figure 3.** Coal spontaneous-combustion process interval stage change situation [33].

### 3.3. Effect of Active Group Content on Gas Products

According to the isothermal temperature difference leading test, the volume fraction of gas products in each group was drawn in a scatter plot, and the gas product data of different extraction residual coal groups were fitted by Logistic using Origin software to obtain the gas product fitting curve (as shown in Figure 4). The absolute production of each gas product was obtained by integrating the fitting curve, and then the correlation between the content of active groups and the concentration of gas products was analyzed to determine the effect of each active group on the concentration of gas products.

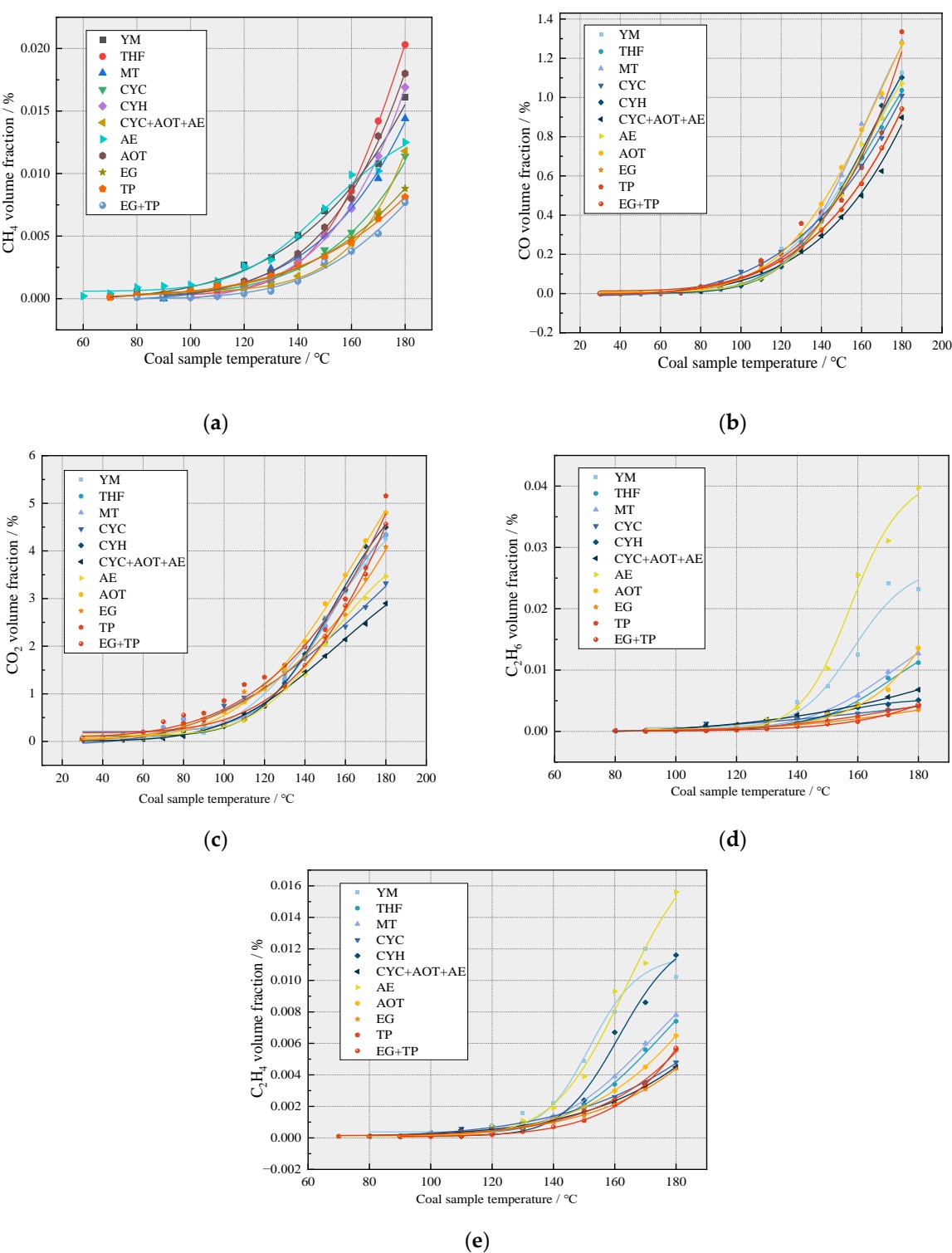

**Figure 4.** Fitting curves of the gas products of different raffinate coal groups [33]. (**a**) Fitting curve of CH$_4$ gas product; (**b**) Fitting curve of CO gas product; (**c**) Fitting curve of CO$_2$ gas product; (**d**) Fitting curve of C$_2$H$_6$ gas product; (**e**) Fitting curve of C$_2$H$_4$ gas product.

As shown in Figure 4, the fitting curves of five gas products of CH$_4$, CO, CO$_2$, C$_2$H$_6$ and C$_2$H$_4$ were obtained, respectively. Origin software was used to integrate the fitting curves, and the absolute production results of each gas product are shown in Table 2.

**Table 2.** The amount of gas products produced by different extraction residual coal groups.

| Group | $CH_4$ | CO | $CO_2$ | $C_2H_6$ | $C_2H_4$ |
|---|---|---|---|---|---|
| YM | 0.48549 | 38.33111 | 177.61669 | 0.62251 | 0.34461 |
| THF | 0.43321 | 35.06035 | 178.00860 | 0.2425 | 0.17216 |
| MT | 0.37073 | 40.99266 | 178.00860 | 0.28744 | 0.18847 |
| CYC | 0.27937 | 37.15417 | 158.69036 | 0.16944 | 0.14322 |
| CYH | 0.37136 | 36.34579 | 174.96191 | 0.1641 | 0.25814 |
| CYC + AOT + AE | 0.24421 | 28.80616 | 127.02185 | 0.23388 | 0.12843 |
| AE | 0.48858 | 36.17914 | 145.47469 | 0.9406 | 0.36273 |
| AOT | 0.43511 | 43.82697 | 201.89459 | 0.23651 | 0.15241 |
| EG | 0.26614 | 32.19166 | 173.87919 | 0.10831 | 0.11619 |
| TP | 0.26061 | 39.30838 | 200.26572 | 0.12891 | 0.13512 |
| EG + TP | 0.18521 | 32.19166 | 169.43187 | 0.09223 | 0.10690 |

Gray Correlation Analysis of Active Groups and Gas Products

The gas product can be used as a sign to divide the process of coal spontaneous combustion, and the change in its content can reflect the activity intensity of coal spontaneous combustion to a certain extent. To explore the relationship between the structure of active groups and the characteristics of coal spontaneous combustion, the content of each active group and the absolute production of different gas products were obtained by infrared peak fitting. The relationship between gas products and the structure of active groups was analyzed by gray correlation analysis, and then the influence of active groups on the characteristics of coal spontaneous combustion was determined [36]. The greater the correlation degree calculated, the greater the influence of the active group structure on the formation of gas products in the process of coal spontaneous combustion. In the process of this analysis, the amount of gas product generation is the reference sequence, and the content of different active groups is the comparison sequence. The results are shown in Figure 5.

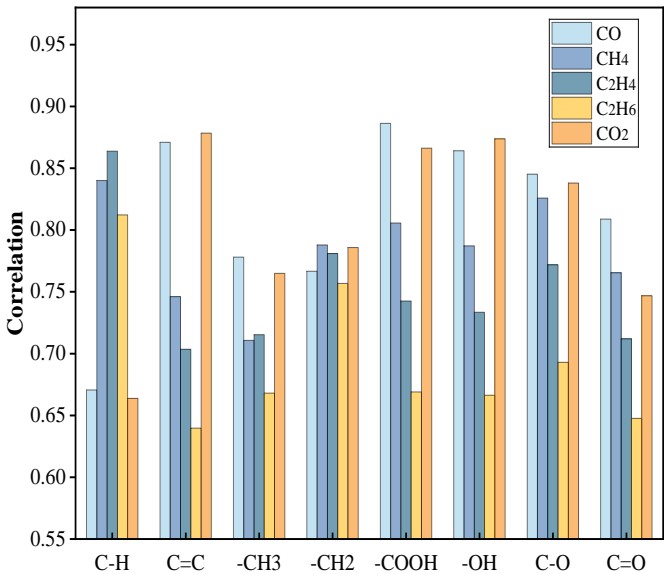

**Figure 5.** Gray correlation degree between active group structure and gas product.

(1) The correlation between CO, $CO_2$ and different active group structures

It can be seen from Figure 5 that the correlation between -COOH, C=C, -OH and the production of CO and $CO_2$ is high, indicating that these three types of active group structures have a great influence on the production of CO and $CO_2$. The correlation degree with CO is 0.8863, 0.8710 and 0.8641, and the correlation degree with $CO_2$ is 0.8661, 0.8783 and 0.8737, respectively. In the early stage of the coal spontaneous-combustion reaction,

-OH as the dominant group captures H and oxidizes to form an aldehyde radical, which further reacts with oxygen to form CO and $CO_2$, resulting in an increase in the production of CO and $CO_2$ gas. In the middle stage of the coal spontaneous-combustion reaction, -COOH combines with $O_2$ to generate CO and $CO_2$ gas, resulting in a continuous increase in CO and $CO_2$ generation. In this stage, -COOH has a great influence on gas production. In the later stage of the coal spontaneous-combustion reaction, the macromolecular benzene ring skeleton (C=C) as the dominant reaction group begins to break and reacts with free oxygen atoms and oxygen, resulting in a continuous increase in CO and $CO_2$ gas production. It can be inferred that a decrease in -COOH, C = C and-OH content will lead to a decrease in CO and $CO_2$ gas production, which will lead to a weakening of coal spontaneous combustion.

(2)    The correlation between $CH_4$, $C_2H_4$, $C_2H_6$ and different active group structures

The results of the Origin fitting show that the production of $CH_4$, $C_2H_4$ and $C_2H_6$ gases is after the middle stage of the coal spontaneous-combustion reaction (60 °C). Therefore, the analysis of the correlation between the active structure and these three gases can reflect the influence of the active group structure on the spontaneous-combustion characteristics of coal after the middle stage of the oxidation reaction. It can be seen from Figure 5 that the groups with a high correlation with $CH_4$, $C_2H_4$ and $C_2H_6$ gases are C-H, -$CH_2$, -COOH and C-O structures. Therefore, it is believed that after the middle stage of coal spontaneous combustion, the oxygen-containing functional groups participate in the reaction, the aliphatic hydrocarbon structure reacts with the oxygen-containing functional groups to generate carbon–oxygen free radicals and further carbonylation will produce hydrocarbon gases such as $CH_4$ and $C_2H_4$. Among them, the active groups with the highest correlation with $CH_4$, $C_2H_4$ and $C_2H_6$ are all C-H structures, and their correlation degrees are 0.8400, 0.8638 and 0.8122, respectively. The C-H structure has the greatest influence on the production of the three gases. At the same time, -$CH_2$, -COOH and C-O all have a greater impact on the production of the three gases, indicating that after the coal spontaneous-combustion reaction enters the medium-term stage, the aromatic structure represented by the C-H structure and the aliphatic hydrocarbon structure represented by -$CH_2$ and some oxygen-containing functional groups have a greater impact on the coal spontaneous-combustion reaction activity. Therefore, it is considered that a decrease in aliphatic hydrocarbon structures such as C-H, -$CH_2$ and -COOH will lead to a decrease in hydrocarbon gas generation such as $CH_4$ and $C_2H_4$, thus weakening the spontaneous-combustion of coal.

### 3.4. Gray Correlation Analysis of Active Groups and Thermal Properties of Coal

The most intuitive characterization of the oxidation activity intensity of coal spontaneous combustion is the weight loss rate and exothermic intensity. Therefore, by determining the ignition-point temperature, heat-release peak temperature and burnout temperature characteristic points in the TG, DTG and DSC data curves, the difference in the coal combustion characteristics under the different spontaneous-combustion intensities of each group of raffinate coal samples is intuitively characterized. The characteristic temperature and corresponding TG data values at each feature point can be determined by processing the thermophysical characterization data of the extracted coal sample. All the determined data are summarized in Table 3.

In the process of this analysis, according to the data obtained in Table 3, the thermal physical parameters of different coals are taken as the reference sequence, and the structure of active groups is taken as the comparison sequence. The gray correlation analysis method is used to calculate the correlation between the structure of different active groups and the thermal properties of coal. The influence of different active group structures on the thermal properties of coal is determined, and then the influence of different active group structures on the spontaneous-combustion characteristics of coal is obtained. The calculation results are shown in Figure 6.

**Table 3.** The characteristic point temperature and corresponding mass value of each group of raffinate coal samples [34].

| Group | Ignition Temperature/°C | Ignition-Point Quality/% | Peak Temperature of Heat Release/°C | Peak Temperature Quality/% | Burning Temperature/°C | Burnout Temperature Quality/% | Spontaneous-Combustion Index Value |
|---|---|---|---|---|---|---|---|
| YM | 402.68 | 93.76 | 459.33 | 68.44 | 543.16 | 23.63 | 628.01 |
| THF | 415.09 | 82.86 | 464.42 | 61.76 | 542.79 | 24.13 | 773.34 |
| MT | 411.51 | 84.76 | 458.10 | 64.63 | 546.38 | 22.88 | 704.32 |
| CYC | 404.66 | 87.17 | 443.06 | 70.81 | 550.84 | 21.43 | 687.42 |
| CYH | 406.93 | 87.31 | 452.94 | 69.26 | 552.29 | 21.95 | 698.24 |
| C + A + A | 417.94 | 75.94 | 453.72 | 62.02 | 543.11 | 21.99 | 790.78 |
| AE | 402.88 | 92.18 | 457.74 | 68.13 | 540.10 | 22.58 | 625.89 |
| AOT | 410.32 | 85.30 | 452.58 | 68.91 | 540.76 | 23.40 | 692.23 |
| EG | 403.72 | 90.14 | 448.07 | 73.16 | 555.70 | 23.90 | 698.63 |
| TP | 400.65 | 89.73 | 454.21 | 72.42 | 545.80 | 23.48 | 712.48 |
| EG + TP | 416.52 | 81.35 | 449.09 | 72.51 | 557.23 | 23.27 | 843.20 |

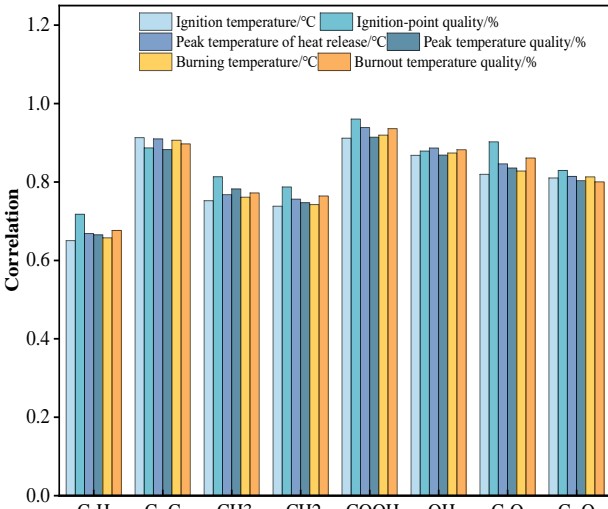

**Figure 6.** Correlation between active group structure and thermophysical parameters.

The ignition-point temperature of coal is the minimum temperature required for the continuous combustion of coal in air, and its size can reflect the strength of coal spontaneous combustion. The calculation results shown in Figure 6 by gray correlation analysis show that C=C, -COOH and -OH have a great correlation with the ignition-point temperature, and the correlation degrees are 0.9132, 0.9116 and 0.8678, respectively. In the process of coal spontaneous combustion, the active group structure reacts with oxygen to release heat. As the dominant groups participating in the reaction, i.e., -COOH, -OH and C=C, the heat released by their oxidation also has a great influence on the ignition temperature. The lower the mass remaining in the combustion stage of coal means the lower the energy intensity released by complete combustion. The analysis results show that the group structures with a great correlation with the quality of the ignition-point are -COOH, C-O and C=C, and their correlation degrees are 0.9604, 0.9026 and 0.8869, respectively. After extraction, the content of -COOH, C-O and C = C groups in coal decreases, resulting in a decrease in the intensity of the coal–oxygen composite reaction and a decrease in the amount of heat released, which is manifested in a decrease in coal quality. In addition, the peak temperature of heat release, the mass of peak temperature, and the mass of burnout temperature reflect the heat-release power of coal after entering the severe combustion stage. It can be seen from Figure 6 that the peak temperature of heat release, the mass of peak temperature, and the mass of burnout temperature have a great correlation with the structure of -COOH, C=C and -OH, indicating that the maximum heat-release power of coal is greatly affected by the structure of -COOH, C=C and -OH.

## 4. Discussion

The influence mechanism of the key active structure of coal molecules based on extraction technology on spontaneous-combustion is summarized by comparing and analyzing the test data by the gray correlation method, as shown in Figure 7.

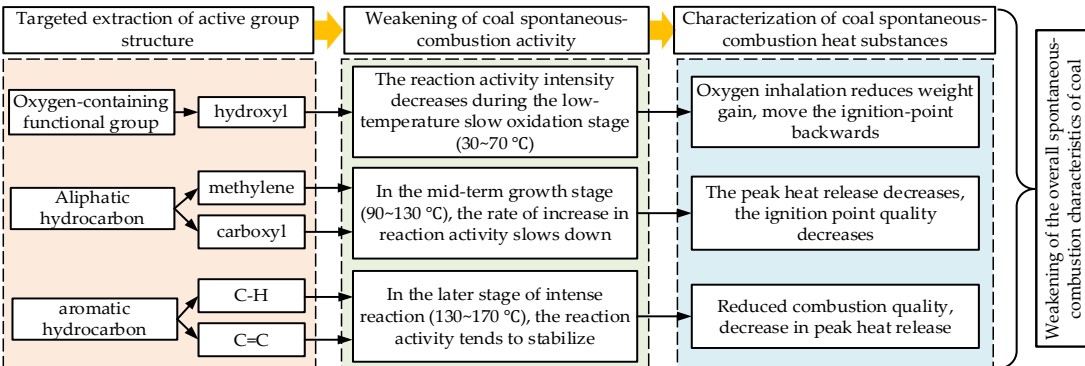

**Figure 7.** The influence mechanism of the key active structure of coal molecules on spontaneous combustion.

It can be seen from Figure 7 that the influence mechanism of the key active structure of coal molecules on spontaneous combustion characteristics is mainly divided into three major links.

(1)    Extraction of the coal molecular active group structure

The most important role in this link is to strip the active group structure of coal molecules through extraction reagents. The essence is to destroy the connection structure between molecules through the chemical action of reagents or the dominant group structure itself in the process of the coal spontaneous-combustion reaction. For example, through various chemical energy modes of action or the characteristic components carried by the reagent, the related connection structure inside the coal molecule is destroyed, thereby stripping some functional group structures from the aromatic main chain of the coal molecules or destroying the macromolecular chain structure. THF reagent materials capture hydrogen bonds through a strong hydrogen bond acceptance ability, while other reagent materials such as pyridine [37], NMP [38] and ionic liquid [39] also have a similar group structure destruction ability.

(2)    Weakening the reactivity intensity of coal spontaneous-combustion

By controlling the structure of the dominant group of the coal spontaneous-combustion reaction from the coal body, the lower limit of the key temperature node of the coal spontaneous-combustion is increased or the temperature range is prolonged, thereby weakening the reactivity of the coal spontaneous combustion in each process stage. For example, the oxygen-containing groups dominated by the -OH structure mainly affect the reactivity of coal spontaneous combustion at low temperature (30~70 °C). The aliphatic hydrocarbon structures represented by -COOH and -CH$_2$ structures mainly affect the reactivity of coal spontaneous combustion in the middle stage (90~130 °C). The aromatic hydrocarbon structures represented by C=C and C-H structures mainly act on the reactivity of coal spontaneous combustion in the later stage (130~170 °C). The effect of the above active group structures on the reactivity of coal spontaneous combustion is manifested in prolonging the temperature range of coal spontaneous combustion or increasing the lower limit of coal spontaneous combustion temperature. After extraction, the content of active structures such as oxygen-containing functional groups, aliphatic hydrocarbons and C=C participating in the oxidation reaction to generate gas is reduced, which leads to a decrease in the production of symbolic gases, and ultimately weakens the spontaneous-combustion of coal.

(3) Weakening the characterization strength of coal thermophysical properties

The microscopic mechanism of coal spontaneous combustion confirms that the essence of coal spontaneous combustion is the process of an oxygen capture reaction and the interaction of active groups in coal molecules, and the ultimate goal of coal spontaneous combustion weakening is to improve the thermophysical characterization strength of the coal oxidation spontaneous-combustion reaction. As the characterization parameters of coal spontaneous combustion activity, ignition-point temperature and ignition-point mass can reflect the size of coal spontaneous combustion reaction activity. By stripping the dominant group of the coal spontaneous-combustion reaction, the ignition temperature of coal will be delayed, and the quality of coal entering the combustion stage will be reduced. It is speculated that the stripping of the active structure will lead to a decrease in heat release and heat-release efficiency in the process of coal oxidation, thus weakening the effect of coal spontaneous combustion.

## 5. Conclusions

According to the previous research results and data of the team, combined with the quantitative indicators of the spontaneous-combustion of each group of raffinate coal, the migration law of spontaneous combustion process nodes and the characterization of thermal properties, the influence of various active group structures on the reaction of the coal spontaneous-combustion process was determined. The main conclusions are as follows:

(1) The influence of the active structure on the process of coal spontaneous combustion is different. The oxygen-containing structure represented by -OH mainly affects the reactivity in the low-temperature oxidation process (30~90 °C). The aliphatic structure represented by -COOH mainly affects the reactivity in the middle stage of coal spontaneous combustion (90~130 °C). The aromatic ring structure determines the reactivity intensity of coal spontaneous combustion in the high-temperature stage (130~170 °C).

(2) The gray correlation method was used to calculate the correlation degree between the active structure of coal molecules and gas products. It was found that the gas products produced in the process of coal spontaneous combustion had a large correlation degree with -OH, -COOH, -CH$_2$, C = C and other structures. With the development of the coal spontaneous-combustion process, various active structures have participated in the reaction, thus generating oxygen-containing gases and alkanes and other iconic gases. The stripping extraction of the active structure leads to a decrease in the content of the groups involved in the reaction, resulting in a decrease in gas production, thereby weakening the spontaneous-combustion of coal.

(3) Through gray correlation analysis, it is found that the main active groups affecting the thermal properties of coal are C=C in the molecular skeleton of coal, -COOH of aliphatic hydrocarbons and oxygen-containing functional groups represented by -OH. In the process of coal spontaneous combustion, the active structure reacts with oxygen to generate heat. The stripping of the active structure leads to a decrease in the intensity of the coal–oxygen composite reaction and a decrease in the amount of heat released, resulting in a delay in the ignition-point temperature and a decrease in the quality of the coal involved in the combustion. Finally, the coal spontaneous-combustion is weakened.

(4) The influence mechanism of the key active structure of coal molecules based on extraction technology on spontaneous combustion characteristics is divided into three parts. (1) Extraction destroys the connection bond between coal molecules or the dominant group structure of the coal spontaneous-combustion reaction. (2) Stripping the dominant group structure of the coal spontaneous-combustion reaction weakens the reaction activity intensity in each process stage of coal spontaneous combustion, prolongs the temperature range of the heat storage reaction or increases the lower limit of the rapid reaction temperature threshold. (3) As a result, the thermal physical

property characterization strength of coal spontaneous combustion is weakened, which is manifested as delaying the ignition-point temperature of coal or reducing the quality of coal involved in combustion.

**Author Contributions:** Conceptualization, supervision, investigation, writing—original draft, J.G.; writing—original draft, writing—review and editing, Y.W.; formal analysis, Y.L.; methodology, investigation, G.C.; methodology, formal analysis, D.L.; software, article proofreading Y.J. All authors have read and agreed to the published version of the manuscript.

**Funding:** This research was supported by the National Natural Science Foundation of China (grant nos. 52004209, 52304251 and 52174198), the Shaanxi Postdoctoral Science Foundation (grant no. 2023BSHEDZZ286) and the Shaanxi Science and Technology Association Young Talents Lifting Project (grant no. 20240205).

**Institutional Review Board Statement:** Not applicable.

**Informed Consent Statement:** Not applicable.

**Data Availability Statement:** Data available on request due to restrictions legal.

**Conflicts of Interest:** Author Yan Jin was employed by the company Taizhou Petroleum Branch, Sinopec Sales Co., Ltd. The remaining authors declare that the research was conducted in the absence of any commercial or financial relationships that could be construed as a potential conflict of interest.

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
