# Peer review of "Study on the Influence Mechanism of the Key Active Structure of Coal Molecules on Spontaneous Combustion Characteristics Based on Extraction Technology"

_fire, doi:10.3390/fire7080283_

Round 1
Reviewer 1 Report
Comments and Suggestions for Authors
The article is well written.
I have noted only a few concerns highlighted below.
Abstract
Your abstract has four sentences. The second and third sentences are too long (i.e., lines 17 to 21 and 21 to 26). This abstract needs to be rewritten to communicate to the reader in clearer ways.
Introduction
The paragraph from lines 40 to 83 can be divided into two or even three paragraphs( e.g., a standalone paragraph on solvent extraction and methods used in analysis).
Results and analysis
Why are the results of the previous studies discussed in Section 3.1? If this is important then there is a a need to have two or three reasons for this in this MS.
Check the spelling of the word structure in line 147
Line 159 to 171: Can you please explain the mechanism of oxygen-containing group extraction? Why YM perform way better than EG+TP?
After going through Sections 3 and 4 I have noted that there is a need to explain mechanisms of the many processes present in these sections.
Comments on the Quality of English LanguageThere is a need to proofread the work and stick to short sentences. Long sentences lose their meaning.
Reviewer 2 Report
Comments and Suggestions for Authors
This paper analyzes the effect of extraction weakening and combines the correlation analysis of various characteristic parameters of spontaneous combustion of extracted coal, and studies the influence mechanism of key active structures of coal molecules on spontaneous combustion characteristics based on extraction technology. Generally, it is well written and can be accepted after the following comments are addressed.
(1) 2.1, why are these extractants selected?
(2) "200 g of samples with five particle sizes were mixed into 1 kg of experimental coal samples". It should be clearly stated what the five particle sizes are.
(3) Table 1. Infrared peak fitting results. what do the numbers in this Table mean? what is the unit?
(4) 2.2. Isothermal temperature difference leading test, what is the heating rate?
English Language should be polished up.
Reviewer 3 Report
Comments and Suggestions for Authors
Dear Authors,
I have carefully studied the article submitted for review and believe that it has high practical and scientific significance. There are a few notes before publishing this work:
1. It is possible to add to the introduction a description of methodologies for studying the influence of the structural and group composition of carbon materials on their properties based on the following works:
- Rudko V.A., Gabdulkhakov R.R., Pyagai I.N. Scientific and technical substantiation of the possibility for the organization of needle coke production in Russia // Journal of Mining Institute. 2023. Vol. 263.p. 795-809. https://pmi.spmi.ru/pmi/article/view/16246
- 10.1016/j.tca.2023.179550
- 10.1016/j.matdes.2023.111952
- 10.1016/S0008-6223(03)00051-4
- 10.1149/1945-7111/acfac4
- 10.1149/1945-7111/ac6a16
- 10.1134/S0036024424030312
2. In section 2.3 it is necessary to describe the operating parameters of the experiment - heating rate, medium, etc.
3. Section 3.2.1 should be in the methods.
4. In conclusion, it is necessary to indicate not only the observed patterns in the influence of functional groups on the combustion process, but also to offer a rational explanation for these patterns.
Reviewer 4 Report
Comments and Suggestions for Authors
No comments
Author Response
Thank you for the recognition of our article. Please see the attachment.
